# ACT-SQL: In-Context Learning for Text-to-SQL with Automatically-Generated Chain-of-Thought

**Hanchong Zhang, Ruisheng Cao, Lu Chen,*  Hongshen Xu and Kai Yu**
X-LANCE Lab, Department of Computer Science and Engineering
MoE Key Lab of Artificial Intelligence, SJTU AI Institute
Shanghai Jiao Tong University, Shanghai, China
{zhanghanchong,chenlusz}@sjtu.edu.cn

## Abstract

Recently Large Language Models (LLMs) have been proven to have strong abilities in various domains and tasks. We study the problem of prompt designing in the text-to-SQL task and attempt to improve the LLMs' reasoning ability when generating SQL queries. Besides the trivial few-shot in-context learning setting, we design our chain-of-thought (CoT) prompt with a similar method to schema linking. We provide a method named ACT-SQL[1] to automatically generate auto-CoT exemplars and thus the whole process doesn't need manual labeling. Our approach is cost-saving since we only use the LLMs' API call once when generating one SQL query. Furthermore, we extend our in-context learning method to the multi-turn text-to-SQL task. The experiment results show that the LLMs' performance can benefit from our ACT-SQL approach. Our approach achieves SOTA performance on the Spider dev set among existing in-context learning approaches.

## 1 Introduction

The text-to-SQL task (Zhong et al., 2017; Xu et al., 2017) aims to translate the natural language question into the corresponding SQL query with the given database schema. It is the key technique to establish the natural language interface on relational databases, which can help common users access data from relational databases in a more convenient way.

Recent studies in text-to-SQL research have primarily centered on the development of semantic parsers within the framework of cross-domain analysis. In cross-domain text-to-SQL datasets such as Spider (Yu et al., 2018), SParC (Yu et al., 2019b), and CoSQL (Yu et al., 2019a), the databases employed in the train set, dev set, and test set do not overlap. Prior research endeavors have focused on training specialized text-to-SQL models and optimizing their structural components to enhance overall performance. Notably, these efforts have yielded impressive model performances across various datasets. Nevertheless, the construction of such models necessitates a substantial number of high-quality training examples and entails significant time investments for finetuning. Moreover, these models often possess intricate structures, rendering their deployment challenging.

Recent research has provided empirical evidence establishing the substantial capabilities of Large Language Models (LLMs), such as GPT-3 (Brown et al., 2020) and ChatGPT (Ouyang et al., 2022), across a wide spectrum of domains and tasks. As the scale of LLMs continues to expand, scholarly investigations have revealed the presence of emergent abilities (Wei et al., 2022) exclusive to larger LLMs and absent in their smaller counterparts. Therefore, the latest studies employ LLMs in the context of the text-to-SQL task, utilizing the in-context learning method (Brown et al., 2020). Owing to the impressive performance demonstrated by LLMs in zero-shot or few-shot prompting scenarios, the need for extensive finetuning using an abundance of training examples has been rendered unnecessary. Consequently, the integration of LLMs in the text-to-SQL process yields notable time and cost savings.

Nonetheless, contemporary in-context learning approaches for text-to-SQL encounter certain challenges. For instance, Rajkumar et al. (2022), in comparison to SOTA finetuned models, employ a simplistic prompt designing approach that yields relatively subpar performance. Similarly, Pourreza and Rafiei (2023) employs a convoluted workflow to generate the final SQL query, resulting in achieving SOTA performance on the test set of the Spider dataset. However, this approach proves time-consuming and resource-intensive, as it ne-

---

*The corresponding author is Lu Chen.

[1]Our code is publicly available at https://github.com/X-LANCE/text2sql-GPT.

cessitates multiple API calls to LLMs during the query generation process. Moreover, the recent advancements in in-context learning methods for text-to-SQL have yet to be extended to multi-turn datasets, such as SParC, CoSQL, and DIR (Li et al., 2023b).

Despite the proficiency of LLMs as zero-shot and few-shot learners, the mere superficial prompt design fails to fully activate their capabilities. To address this limitation, Wei et al. (2023) proposes a novel prompting technique called chain-of-thought (CoT). Through the CoT method, the prompt text encompasses a comprehensive thinking process that guides LLMs towards accurate deduction of answers. Notably, the CoT method mirrors the sequential nature of human reasoning, wherein intermediate answers are obtained before arriving at a final conclusion. Given the intricate nature of the text-to-SQL task, the CoT method proves highly suitable, as generating the SQL query entails complex reasoning processes. However, existing CoT methodologies necessitate extensive time investments in the selection of canonical examples and manual labeling. The text-to-SQL task lacks an automated approach for generating CoT sequences.

In this paper, we propose our in-context learning method for the text-to-SQL task with the automatically-generated CoT. First, under the zero-shot setting, we study the influence on LLMs' performance caused by the input format of the database schema. Second, under the few-shot setting, we provide a hybrid strategy to select exemplars and study the influence on LLMs' performance caused by the number of exemplars. Our experiment results show that the strategy is effective. Third, we present our approach named **ACT-SQL** to generate auto-CoT for the dataset training example consisting of the database schema, the question, and the corresponding SQL query. The experiment results show that the generated auto-CoT can indeed improve the LLMs' performance. The ACT-SQL achieves the SOTA performance on the Spider dev set among existing in-context learning methods. In addition, the ACT-SQL does not need to use extra LLMs' API calls, which means that our workflow is relatively fast and cheap. Finally, we apply our approach in multi-turn text-to-SQL datasets including SParC and CoSQL and achieve comparable accuracy scores with finetuned models. Our main contributions can be summarized as follows:

1. We explore the influence on LLMs' perfor-

mance under the text-to-SQL task with different prompting styles and few-shot exemplar selection strategies.

2. We propose our approach named **ACT-SQL** to generate auto-CoT. The ACT-SQL achieves the SOTA performance on the Spider dev set among existing in-context learning methods. Furthermore, our automatic method is cost-saving and time-saving and does not need extra LLMs' API calls.

3. We extend our method onto the multi-turn text-to-SQL task and achieve comparable performances with finetuned models on the SParC and CoSQL datasets.

## 2 Related Work

**Text-to-SQL models** Over the past several years, text-to-SQL researches mainly focus on building well-designed deep neural networks (Chen et al., 2021b; Cao et al., 2023). RATSQL model (Wang et al., 2020) and LGESQL model (Cao et al., 2021) are AST-based approaches, where AST is the abbreviation of the abstract syntax tree. They encode the input and decode the AST of the SQL query with predefined grammar. AST-based approaches perform well but are generally complex to deploy.

PICARD (Scholak et al., 2021) is a sequence-to-sequence model. SQL is a formal language that follows strict grammar rules. Directly finetuning pretrained language models (PLMs) on text-to-SQL datasets would make PLMs likely to generate invalid SQL queries. The PICARD model rejects invalid tokens at each decoding step and constrains the generated results into a certain output space.

Although these specialized models have achieved excellent performances, there still exist some inevitable disadvantages. In order to train a text-to-SQL model, abundant high-quality training examples are needed. Constructing and labeling a large-scale text-to-SQL dataset is always not easy and would consume a lot of resources and time. Training and finetuning the model is also a hard project which costs many computing resources.

**In-context learning for text-to-SQL** Since LLMs have shown amazing ability across various domains and have been applied in many academic and industrial fields, the latest researches begin to activate the LLMs' ability for the text-to-SQL task. Rajkumar et al. (2022) uses the trivial zero-shot and few-shot learning setting and performs an

empirical evaluation of text-to-SQL capabilities of LLMs including GPT-3 (Brown et al., 2020) and Codex (Chen et al., 2021a). They perform the zero-shot prompt learning on Spider (Yu et al., 2018), a large-scale human-labeled cross-domain text-to-SQL dataset. Their work is relatively simple and the performance falls behind finetuned models.

Nan et al. (2023) mainly concentrates on the strategy of exemplars selection. Their work achieves good performance on several cross-domain datasets including Spider, Spider-Syn (Gan et al., 2021a), Spider-DK (Gan et al., 2021b) and Spider-Realistic (Deng et al., 2021). However, their work requires an extra preliminary predictor to evaluate the SQL's difficulty level and needs to use LLMs' API call many times due to the majority vote method.

DIN-SQL (Pourreza and Rafiei, 2023) provides a relatively complex approach. DIN-SQL consists of a complex workflow that decomposes the problem into several simpler sub-problems. With the LLM GPT-4, DIN-SQL has surpassed previous finetuned models and has achieved the best score on the Spider dataset. But DIN-SQL's workflow is obviously slow and expensive since it uses LLMs' API call many times to generate one SQL.

## 3   Methodology

With the in-context learning method, the SQL generation process can be formulated as

$$S = \text{LLM}(I, D, Q, \mathcal{E}).$$

$I$ represents the instruction. $D$ represents the database schema. $Q$ represents the question. $\mathcal{E} = [(D_1, Q_1, P_1), \cdots, (D_n, Q_n, P_n)]$ is the list of exemplars where $P_i$ is the answer prompt which contains the correct SQL for the $i$-th exemplar. Thus the performance of LLMs is mainly influenced by the database prompt style, the exemplar selection strategy, and the exemplar prompt design.

In this section, we first describe the prompt styles of the database schema. Then we state our strategy of exemplar selection for the few-shot learning setting. Furthermore, we introduce our ACT-SQL approach, i.e. the automatically generated CoT method for constructing effective answer prompts. Finally, we extend our approach to the multi-turn text-to-SQL task.

### 3.1   Database Prompt Style

Previous works have shown that given the database schema, strong LLMs (e.g. GPT models) can translate the relatively simple natural language question into the correct SQL query, though no exemplar is provided. Under the zero-shot setting, the LLMs merely take the database schema and the question as the input. Thus the input format of the database schema would mainly influence the LLMs' performance. Generally, we use five different database schema styles, which are shown in Appendix C.1:

1. `Table(Column)` lists each table followed by its columns in each line. This style follows the official document provided by OpenAI[2].

2. `Table(Column)(PF)` adds primary keys and foreign keys at the end of `Table(Column)`.

3. `Create(NoPF)` describes all tables and columns with the "create table" statement in the SQL grammar. "NoPF" represents that no information on primary keys and foreign keys is added. Compared with `Table(Column)`, this input format contains information on column types (e.g. number and text) and is more similar to real SQL statements.

4. `Create(EoC)` adds primary keys and foreign keys based on `Create(NoPF)`. "EoC" represents that they are added at the end of the corresponding column.

5. `Create(EoT)` adds primary keys and foreign keys based on `Create(NoPF)`. "EoT" represents that they are added at the end of the table.

Furthermore, database contents are concerned. Specifically $c$ example rows are appended to each table. Appendix C.2 shows instances where $c = 3$.

### 3.2   Exemplar Selection

Given a few exemplars, LLMs can benefit and acquire tips from them and thus generate SQL queries with a more standard format and higher accuracy. Exemplar selection is an important work under the few-shot setting, which would influence the LLMs' performance a lot.

We select exemplars using a hybrid strategy. Specifically, we first of all select $n_s$ examples from the training dataset at random. These dataset examples are named static exemplars. They would be used in the context of every test case. As for each specific test case, we select $n_d$ extra examples from

---

[2]https://platform.openai.com/examples/default-sql-translate

the training dataset. These dataset examples are named dynamic exemplars since they are selected according to some features of the current test case. Consequently, there are total $n_s + n_d$ exemplars for each test case.

In order to get dynamic exemplars that are more relevant to the current test case, we compare the natural language question of the current test case with all questions in the training dataset. We calculate the similarity scores with the suitable pretrained model and then select the top-$n_d$ training dataset examples. We believe that dynamic exemplars with more relevant questions would provide more effective information to the LLMs.

### 3.3 Chain-of-Thought Prompt Design

Under the few-shot learning setting, it has been proven that the LLMs' performance can benefit a lot from the chain-of-thought (CoT) (Wei et al., 2023) method. In the text-to-SQL task, only the database schema, the question, and the corresponding SQL query are provided in the prompt under the trivial few-shot learning setting. However, with the CoT method, the thought process of how to write the correct SQL query is added to the prompt. These prompting texts can help the LLMs think step by step when generating the complete SQL query and thus can activate the logical reasoning ability of the LLMs.

In previous works, some grammar-based text-to-SQL models utilize the graph encoding technique to jointly encode both the database schema and the question. Schema linking (Bogin et al., 2019; Wang et al., 2020; Cao et al., 2021) is a commonly used algorithm for building the input graph. If the question tokens exactly or partially match some schema item (i.e. table and column) names, then they are linked with the specific graph edge. It is obvious that the schema linking method can help the text-to-SQL models fetch the most relevant tables and columns among a great number of schema items based on the question.

**DB ID:** store_product
**Question:** Find the name all districts with city area greater than 10 or population larger than 100000.
**SQL:** SELECT District_name FROM district WHERE City_Area > 10 OR City_Population > 100000
**CoT:**
Let's think step by step.
According to "name all districts", columns [district.District_name] may be used.
According to "city area", columns [district.City_Area] may be used.
According to "population", columns [district.City_Population] may be used.
Values [10, 100000] may be used.
So the final answer is:
SELECT District_name FROM district WHERE City_Area > 10 OR City_Population > 100000

Figure 1: Manually labeled CoT for the dataset example.

We design our chain-of-thought prompt with a similar method to schema linking. Figure 1 shows an instance of the manually labeled CoT for the example from the train set of the Spider dataset (Yu et al., 2018). As suggested in Kojima et al. (2023), the CoT prompt starts with "Let's think step by step". For each slice of the question sentence that may contain some information about the schema item, we add them into the CoT prompting text in the format shown in Figure 1. Furthermore, the values mentioned in the question and the SQL query are also a concern. The final SQL query is appended at the end of the CoT prompt.

**Auto-CoT** Although CoT prompts can be manually labeled, it costs a lot of time to find sufficient canonical and effective training dataset examples for CoT labeling. In addition, manually labeled CoT exemplars are fixed, which means that they are all static exemplars and dynamic exemplars are deficient. In order to deal with problems in the manual labeling process, we introduce an automatic method to generate auto-CoT prompts for every example in the training dataset.

Given the question $q = (q_1, q_2, \cdots, q_{|q|})$ and the SQL query $s$, the $q_i$ represents the $i$-th token in the question sentence. We define $q_{i,j} = (q_i, q_{i+1}, \cdots, q_j)$ as a slice of the original question. We first enumerate each column $[tab].[col]$ appearing in the SQL query, where $[tab]$ represents the table name and $[col]$ represents the column name. For each column, we use the suitable pretrained model to compute the similarity scores between the current column and all the question sentence slices. The most relevant slice is

$$\underset{q_{i,j}}{\arg\max} \ \mathrm{Sim}([tab].[col], q_{i,j}),$$

where $\mathrm{Sim}$ is the similarity function. We link the column and its most relevant slice and add them to the auto-CoT prompt in the same format as the manual labeled CoT prompt. Note that during this process, we ignore the column appearing in the GROUP BY clause of the SQL query, since the GROUP BY column is commonly not mentioned directly in the question.

Secondly, we enumerate each table $[tab]$ appearing in the SQL query, where $[tab]$ represents the table name. In this process, we eliminate tables that have occurred in the columns, since those tables have been added into the auto CoT prompt. The left tables only appear in the FROM clause and

indicate some extra information. For each table, we also compute all the similarity scores and find out the most relevant question slice, i.e.,

$$\arg\max_{q_{i,j}} \; \mathrm{Sim}([tab], q_{i,j}).$$

We link the table and its most relevant slice and add them to the auto-CoT.

Finally, we enumerate the values in the SQL query and then add them to the auto-CoT. Figure 2 shows an instance of the auto-generated CoT from the train set of the Spider dataset.

**Auto-CoT**

**DB ID:** twitter_1
**Question:** What is the partition id of the user named "Iron Man".
**SQL:** SELECT partitionid FROM user_profiles WHERE name = 'Iron Man'
**CoT:**
Let's think step by step.
According to "user named", columns [user_profiles.name] may be used.
According to "partition id of the user", columns [user_profiles.partitionid] may be used.
Values [Iron Man] may be used.
So the final answer is:
SELECT partitionid FROM user_profiles WHERE name = 'Iron Man'

Figure 2: Auto-CoT for the dataset example.

### 3.4 Extension for Multi-turn Text-to-SQL

The prompts described in the previous sections are designed for the single-turn text-to-SQL task. However, questions in the multi-turn text-to-SQL task are context-dependent and thus those prompts cannot be directly used. Moreover, the auto-CoT method is also disabled under the multi-turn setting, since the auto-CoT method finds information about schema linking based on the question slices. Under the multi-turn setting, this information may distribute into several context-dependent sentences.

In order to deal with the challenge of the multi-turn text-to-SQL task, we use LLMs to convert the multi-turn text-to-SQL task into the single-turn text-to-SQL task. Concretely, with the help of the LLMs, we can rewrite the question sentences and remove the context dependency among them. Thus each rewritten question and its corresponding SQL query turn into a new independent dataset example. We then directly apply the previous in-context learning method in the converted multi-turn text-to-SQL task.

The quality of the rewritten questions would influence the LLMs' performance a lot. It is necessary to manually label some rewriting exemplars in order to fix the format and improve the quality of the LLMs' outputs. For each multi-turn text-to-SQL dataset, we select 10 examples from the train set at random and manually label the rewritten results.

## 4 Experiments

### 4.1 Experiment Setup

**Models**  We mainly use the GPT-3.5-turbo model to evaluate our proposed approach. The GPT-3.5-turbo model is a low-cost LLM and is very large to have the emergent ability (Wei et al., 2022) for handling the text-to-SQL task. In addition, we use the GPT-4 model to evaluate our auto-CoT method on the Spider dataset (Yu et al., 2018), since the GPT-4 model has a stronger reasoning ability but is much more expensive. We use the PLM text2vec-base-chinese to compute the similarity scores when selecting dynamic exemplars and generating auto-CoT prompts.

**Hyperparameters**  The temperature in LLMs' API is set to 0, i.e. the greedy decoding strategy is applied. The text-to-SQL tasks require the model to generate SQL queries with strict grammar rules. The LLMs are likely to generate invalid SQL queries or to write SQL queries that are not relevant to the given questions if the temperature is too high. The number of max tokens is set to 150 for the trivial in-context learning setting and 750 when using the CoT method.

**Datasets**  We mainly evaluate our proposed approach on Spider, a large-scale human-labeled cross-domain text-to-SQL dataset across 200 databases covering 138 domains. The Spider dataset contains 8,659 examples in the train set and 1,034 examples in the dev set. It also provides the evaluation script which divides SQL queries into four categories (i.e. easy, medium, hard, and extra) according to the difficulty level. The test set of Spider is not publicly available. We conduct the experiments on the dev set.

In addition, we also conduct the in-context learning experiments on Spider-Syn (Gan et al., 2021a), Spider-DK (Gan et al., 2021b) and Spider-Realistic (Deng et al., 2021). Based on Spider, Spider-Syn replaces some schema-related tokens in the question with synonyms, which would make models unable to discover useful schema items with the simple string-matching method. Spider-DK defines five types of domain knowledge and modifies some examples by adding domain knowledge that reflects real-world question paraphrases. Spider-DK evaluates the models' generalization ability across domains when domain knowledge does not frequently appear in the train set. Spider-Realistic removes explicit mentions of column

names to evaluate the model's ability to capture text-table alignment.

As for multi-turn text-to-SQL datasets, we conduct our experiments on SParC (Yu et al., 2019b) and CoSQL (Yu et al., 2019a). SParC totally consists of 4,298 coherent question sequences including 12k+ individual questions and the corresponding SQL queries. CoSQL totally contains 10k+ annotated SQL queries. Each dialogue in CoSQL simulates a real-world scenario where the common user is exploring the database and the expert is retrieving answers with SQL.

**Evaluation metrics** We use three commonly used evaluation metrics of the text-to-SQL task: exact match accuracy (EM), execution accuracy (EX), and test-suite accuracy (TS). The EM metric requires each component of the predicted SQL to be equivalent to the corresponding component of the gold SQL. Values in the SQL query are not concerned with the EM metric. The EX metric requires the execution result of the predicted SQL to be correct. Since there may exist different SQL queries that represent the same semantic, the EX metric is commonly more precise than the EM metric. The TS metric also evaluates the execution result but requires the result to be correct under multiple database instances per database schema[3].

For multi-turn text-to-SQL datasets, we evaluate our approach with question match accuracy (QM) and interaction match accuracy (IM). The QM score is 1 if the predicted SQL query for the single question is correct. The IM score is 1 if all the predicted SQL queries in the interaction are correct.

## 4.2 Zero-shot Results

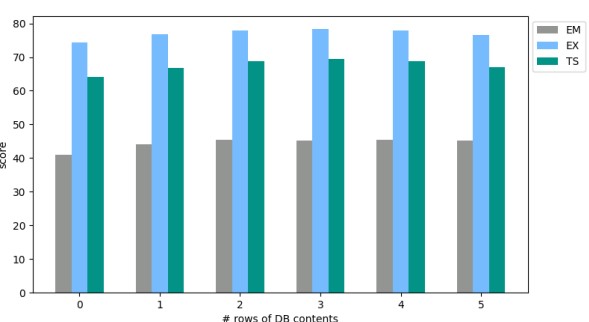

Figure 3: Zero-shot performances of GPT-3.5-turbo using `Table(Column)` DB style with different rows of DB contents on Spider dev set.

[3]https://github.com/taoyds/test-suite-sql-eval

As discussed in Section 3.1, the LLMs' performance is mainly influenced by the database prompt style and the rows of database contents under the zero-shot learning setting. We first conduct experiments for studying the influence on LLMs' performance caused by the rows of database contents. We fix the LLM as the GPT-3.5-turbo model and the database style as `Table(Column)` and only change the rows of database contents for each table in the prompt. Figure 3 shows the result on the Spider dev set. The LLM achieves the lowest score when no database content is provided. This indicates that database contents can provide useful tips for LLMs, especially when the testing case is sensitive to values in SQL where Table 1 shows two cases. In the first case, the 3 records from the database contain exactly one cell value "France" instead of "French" for the column "singer.Citizenship". Thus the LLM successfully predicts the correct value when these records are added to the prompt. In the second case, the database contents point out that "Aberdeen" is the city name so that the LLM can predict the correct SQL structure.

| |
|---|
| **Question:** *What are the names of the singers who are not French citizens?* |
| **DB content 0:** SELECT Name FROM singer WHERE Citizenship != 'French' |
| **DB content 3:** SELECT Name FROM singer WHERE Citizenship != 'France' |
| **Questions:** *Give the flight numbers of flights leaving from Aberdeen.* |
| **DB content 0:** SELECT FlightNo FROM flights WHERE SourceAirport = 'Aberdeen' |
| **DB content 3:** SELECT FlightNo FROM flights WHERE SourceAirport IN (SELECT AirportCode FROM airports WHERE City = 'Aberdeen') |

Table 1: Case study for different rows of DB contents.

The LLM gets the best score when the rows of database contents is set to 3. Too much database content in the prompt would not improve the LLMs' performance. Therefore, we always set the rows of database contents to 3 for the subsequent experiments.

Table 3 shows the different performances of the GPT-3.5-turbo model when using different database styles. In general, `Table(Column)` and `Table(Column)(PF)` achieve higher scores than the other three database styles with the zero-shot learning setting because these two database styles follow the OpenAI's official document and

| Finetuned Model | EM | EX | TS |
|---|---|---|---|
| T5-3B+PICARD (Scholak et al., 2021) | 75.5 | 79.3 | 71.9 |
| RASAT+PICARD (Qi et al., 2022) | 75.3 | 80.5 | - |
| N-best List Rerankers + PICARD (Zeng et al., 2022) | 76.4 | 80.6 | - |
| Graphix-3B+PICARD (Li et al., 2023c) | 77.1 | 81.0 | - |
| RESDSQL-3B+NatSQL (Li et al., 2023a) | **80.5** | **84.1** | - |

| LLM | Approach | API per SQL | EM | EX | TS |
|---|---|---|---|---|---|
| Codex Davinci | Rajkumar et al. (2022) | 1 | - | 67.0 | 55.1 |
| Codex Davinci | Chang and Fosler-Lussier (2023) | 1 | - | 76.8 | - |
| Codex Davinci | DIN-SQL (Pourreza and Rafiei, 2023) | 4 | 57.2 | - | 69.9 |
| GPT-4 | DIN-SQL (Pourreza and Rafiei, 2023) | 4 | 60.1 | 82.8 | 74.2 |
| GPT-3.5-turbo | **ACT-SQL (Ours)** | 1 | **62.7** | 80.4 | 71.4 |
| GPT-4 | **ACT-SQL (Ours)** | 1 | 61.7 | **82.9** | **74.5** |

Table 2: Performances of ACT-SQL and other previous works on Spider dev set.

| DB Style | EM | EX | TS |
|---|---|---|---|
| Table(Column) | 45.3 | 78.3 | **69.4** |
| Table(Column)(PF) | **45.4** | 79.0 | 69.1 |
| Create(NoPF) | 45.3 | 77.0 | 66.1 |
| Create(EoC) | 44.8 | **79.2** | 67.7 |
| Create(EoT) | 44.8 | 78.3 | 67.9 |

Table 3: Zero-shot performances of GPT-3.5-turbo with different DB styles on Spider dev set.

may be more similar to LLMs' pretrained data. `Create(EoC)` and `Create(EoT)` performs better than `Create(NoPF)` in EX and TS metrics. This indicates that primary keys and foreign keys in the prompt can offer LLMs effective information.

### 4.3 Few-shot Results

Table 11 shows all the few-shot experiment results on the Spider dev set, where different database styles and different numbers of static and dynamic exemplars are used. Compared with the zero-shot results, it is obvious that all the EM scores increase a lot. This is because SQL queries from the same dataset usually share similar grammar and structure and thus the exemplars from the Spider train set lead LLMs to output a similar SQL query.

Under the trivial few-shot learning setting, the TS scores also get improved by 1%-3% except for the `Table(Column)` database style. `Table(Column)` no longer performs better than `Table(Column)(PF)`, since LLMs' accuracy for predicting hard and extra hard SQL queries get increased with the few-shot exemplars and thus primary keys and foreign keys in the prompt become more important.

The experiment results prove that our ACT-SQL approach is effective. When the GPT-3.5-turbo model uses the ACT-SQL approach with the `Create(EoT)` database style, it achieves the best EM score of 62.7% and the best TS score of 71.4%. The best database style changes because LLMs can learn from exemplars. Table 13 shows the case study for the ACT-SQL method. With the trivial few-shot learning setting, there is a redundant column "TV_Channel.Hight_definition_TV" appearing in the SELECT clause. When the ACT-SQL method is applied, the entire output generated by the LLM contains the complete thinking process which successfully does the schema linking. After clarifying all the tables and columns that may be used in SQL, the LLM eventually writes the correct SQL query without any redundant schema item.

Since the GPT-4 model is expensive, we use the GPT-4 model to evaluate our ACT-SQL approach only with the `Create(EoT)` database style and $n_s = n_d = 2$. Table 2 shows the performances of our ACT-SQL and other previous works using in-context learning with LLMs. The ACT-SQL approach uses the LLMs' API call only once for generating one SQL query and achieves the highest EM, EX, and TS scores among existing in-context learning approaches. ACT-SQL's performance is also comparable to finetuned models. Actually, finetuned models would get higher scores on the dev set than the test set, since these models are selected by the dev set performance. Instead, in-context learning methods would not suffer the performance gap between the dev set and the test set. Table 4 shows some previous works' performances on Spider dev set and test set. For finetuned ap-

proaches mentioned in the table, the performances drop from the dev set to the test set. On the contrary, for in-context learning approaches mentioned in the table, the performances increase from the dev set to the test set. After all, finetuned models are selected by the dev set performance, which would lead to the overfitting on the dev set and the performance dropping on the test set. For in-context learning approaches, the dev set and the test set are equal to the model. Performances between the dev set and the test set are only affected by the dataset feature.

| Finetuned Approach | Dev | Test |
|---|---|---|
| Graphix-3B+PICARD | 81.0 | 77.6 |
| RESDSQL-3B+NatSQL | 84.1 | 79.9 |
| **In-context Learning** | **Dev** | **Test** |
| C3 (Dong et al., 2023) | 81.8 | 82.3 |
| DIN-SQL (Pourreza and Rafiei, 2023) | 82.8 | 85.3 |

Table 4: Performances of different previous approaches on Spider dev set and test set.

Table 5, Table 6 and Table 7 shows the GPT-3.5-turbo's performances on Spider-Syn, Spider-DK, and Spider-Realistic dev set. We use the Create(EoT) database style and set $n_s = n_d = 2$. The experiment results show that our approach is still comparable to finetuned models on Spider-Syn and Spider-Realistic datasets. On the Spider-DK dataset, our approach's EX score surpasses finetuned models. This is due to the wide range of domain knowledge stored in LLMs.

| Approach | EM | EX | TS |
|---|---|---|---|
| Graphix-3B+PICARD | 66.9 | - | - |
| RESDSQL-3B+NatSQL | 69.1 | 76.9 | - |
| Few-shot (Ours) | 47.2 | 63.7 | 54.5 |
| ACT-SQL (Ours) | 51.5 | 67.9 | 59.3 |

Table 5: Performances of GPT-3.5-turbo and other previous works on Spider-Syn dev set.

| Approach | EM | EX | TS |
|---|---|---|---|
| Graphix-3B+PICARD | 51.2 | - | - |
| RESDSQL-3B+NatSQL | 53.3 | 66.0 | - |
| Few-shot (Ours) | 50.7 | 68.8 | 61.7 |
| ACT-SQL (Ours) | 49.5 | 68.2 | 62.4 |

Table 6: Performances of GPT-3.5-turbo and other previous works on Spider-DK dev set.

| Approach | EM | EX | TS |
|---|---|---|---|
| Graphix-3B+PICARD | 72.4 | - | - |
| RESDSQL-3B+NatSQL | 77.4 | 81.9 | - |
| Few-shot (Ours) | 52.4 | 76.4 | 62.0 |
| ACT-SQL (Ours) | 53.5 | 75.8 | 61.2 |

Table 7: Performances of GPT-3.5-turbo and other previous works on Spider-Realistic dev set.

| Approach | QM | | | IM | | |
|---|---|---|---|---|---|---|
| | EM | EX | TS | EM | EX | TS |
| GAZP+BERT (Zhong et al., 2020) | 48.9 | 47.8 | - | - | - | - |
| RASAT+PICARD (Qi et al., 2022) | 67.7 | 73.3 | - | 49.1 | 54.0 | - |
| Few-shot (Ours) | 48.4 | 64.0 | 55.8 | 24.6 | 39.8 | 31.5 |
| ACT-SQL (Ours) | 51.0 | 63.8 | 56.9 | 24.4 | 38.9 | 29.6 |

Table 8: Performances of GPT-3.5-turbo and other previous works on SParC dev set.

## 4.4 Multi-turn Datasets Results

Table 8 and Table 9 show the GPT-3.5-turbo's performances on two multi-turn text-to-SQL datasets, i.e. SParC and CoSQL. The database style is set to Create(CoT) and $n_d, n_s$ are set to 2 as before. The ACT-SQL approach is not that effective when applied to multi-turn datasets. We believe that our two-phase method causes bad performance. In the first phase, we use LLMs to rewrite questions in the interaction and convert the multi-turn dataset into the single-turn dataset. Sometimes the rewritten result's quality is bad, which influences the schema-linking process. Table 10 shows two rewritten instances from the SParC dev set. In the first instance, the LLM correctly rewrites all sentences without missing any key information. However, in the second instance, the LLM does not remove the context dependency for the second sentence. This also leads to the error in the third sentence, where the keyword "airline" in the first sentence is missing. In general, our in-context learning method is comparable to finetuned models (GAZP + BERT) though there is still a big room for improvement. Improving LLMs' performance on this difficult task is a challenging future work. We just complete the initial exploration.

| Approach | QM | | | IM | | |
|---|---|---|---|---|---|---|
| | EM | EX | TS | EM | EX | TS |
| GAZP+BERT | 42.0 | 38.8 | - | - | - | - |
| RASAT+PICARD | 58.8 | 67.0 | - | 27.0 | 39.6 | - |
| Few-shot (Ours) | 45.4 | 63.9 | 55.5 | 14.7 | 29.4 | 22.9 |
| ACT-SQL (Ours) | 46.0 | 63.7 | 55.2 | 13.3 | 30.7 | 21.5 |

Table 9: Performances of GPT-3.5-turbo and other previous works on CoSQL dev set.

**Original questions:**

1. What are all the flights that leave from Aberdeen?
2. Of those, which land in Ashley?
3. How many are there?

**Rewritten questions:**

1. What are all the flights that depart from Aberdeen? ✓
2. What are the flights that depart from Aberdeen and land in Ashley? ✓
3. How many flights depart from Aberdeen and land in Ashley? ✓

- - - - - - - - - - - - - - - - - - - - - - - -

**Original questions:**

1. What are all the airlines?
2. Of these, which is Jetblue Airways?
3. What is the country corresponding it?

**Rewritten questions:**

1. What are all the airlines? ✓
2. Which one of them is Jetblue Airways? ✘
3. What is the country corresponding to Jetblue Airways? ✘

Table 10: Case study for rewritten questions from SParC dev set.

## 5 Conclusion

LLMs have shown a strong ability in various domains with the in-context learning method. The latest studies have attempted to use LLMs to solve the text-to-SQL task. However, previous prompting approaches either perform worse than finetuned models or need to use LLMs' API call many times. We design the CoT prompt which can be automatically generated and propose our ACT-SQL approach. The ACT-SQL approach uses LLMs' API call only once to generate one SQL query. The experiment results prove that our approach achieves state-of-the-art performance on the Spider dev set among existing in-context learning approaches. Furthermore, we extend our approach to multi-turn text-to-SQL datasets.

## Limitations

There are some limitations in our work. First of all, we use a hybrid strategy for the exemplar selection. The numbers of static and dynamic exemplars are hyperparameters and still need manually determined. In addition, it is a relatively simple strategy that still needs improvement. Furthermore, our approach achieves relatively poor scores on some robustness variants of the Spider dataset and some multi-turn text-to-SQL datasets. Exploration of these datasets can be conducted in future work.

## Acknowledgements

We thank all the anonymous reviewers for their thoughtful comments. This work has been supported by the China NSFC Project (No.62106142 and No.62120106006), Shanghai Municipal Science and Technology Major Project (2021SHZDZX0102), and Startup Fund for Young-man Research at SJTU (SFYR at SJTU).

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

# A   Detailed Experiment Results

| DB Style | $n_s$ | $n_d$ | Few-shot | | | ACT-SQL | | |
|---|---|---|---|---|---|---|---|---|
| | | | **EM** | **EX** | **TS** | **EM** | **EX** | **TS** |
| Table(Column) | 0 | 4 | 58.0 | 77.5 | 68.8 | 58.3 | 78.9 | 69.2 |
| | 2 | 2 | 59.7 | 77.9 | 68.7 | 60.7 | 78.7 | 69.4 |
| | 4 | 0 | 55.1 | 78.6 | 68.6 | 55.9 | 78.1 | 69.4 |
| Table(Column)(PF) | 0 | 4 | 59.3 | 79.9 | 70.6 | 60.8 | 79.8 | 70.2 |
| | 2 | 2 | 60.1 | 79.1 | 69.3 | 62.3 | 79.1 | 71.0 |
| | 4 | 0 | 54.8 | 80.0 | 70.3 | 57.0 | 77.9 | 69.6 |
| Create(NoPF) | 0 | 4 | 58.8 | 78.2 | 68.8 | 58.8 | 79.8 | 68.7 |
| | 2 | 2 | 60.0 | 79.2 | 68.9 | 60.6 | 79.1 | 69.1 |
| | 4 | 0 | 56.3 | 79.4 | 69.8 | 57.4 | 80.7 | 70.7 |
| Create(EoC) | 0 | 4 | 58.5 | 79.5 | 69.0 | 58.6 | 78.7 | 68.7 |
| | 2 | 2 | 60.1 | 79.4 | 68.7 | 61.9 | 79.0 | 71.1 |
| | 4 | 0 | 55.1 | 79.5 | 68.9 | 58.1 | **80.8** | 71.0 |
| Create(EoT) | 0 | 4 | 59.4 | 79.3 | 69.5 | 60.3 | 79.1 | 69.8 |
| | 2 | 2 | 61.4 | 79.4 | 69.0 | **62.7** | 80.4 | **71.4** |
| | 4 | 0 | 56.6 | 80.1 | 69.6 | 55.5 | 78.0 | 68.5 |

Table 11: Few-shot performances of GPT-3.5-turbo with different DB styles and different numbers of static and dynamic exemplars on Spider dev set.

| LLM | Approach | Easy | Medium | Hard | Extra | All |
|---|---|---|---|---|---|---|
| GPT-3.5-turbo | ACT-SQL (Ours) | 89.1 | 77.8 | 62.6 | 36.7 | 71.4 |
| GPT-4 | ACT-SQL (Ours) | 91.1 | 79.4 | 67.2 | 44.0 | 74.5 |

Table 12: Detailed performances of ACT-SQL on Spider dev set.

# B   Auto-CoT by LLM

**Question:** *Find the package choice and series name of the TV channel that has high definition TV.*
**Few-shot:** SELECT series_name, Hight_definition_TV, Package_Option FROM TV_Channel WHERE Hight_definition_TV = 'yes'
**ACT-SQL:** SELECT package_option , series_name FROM TV_Channel WHERE Hight_definition_TV = 'yes'
**Entire Output:**
Let's think step by step.
According to "TV channel that has high definition TV", columns [TV_Channel.Hight_definition_TV] may be used.
According to "package choice and series name", columns [TV_Channel.Package_Option] and [TV_Channel.series_name] may be used.
So the final answer is:
SELECT package_option , series_name FROM TV_Channel WHERE Hight_definition_TV = 'yes'

Table 13: Case study for ACT-SQL. The entire output is totally generated by the LLM with the ACT-SQL approach.

# C   Prompts

In this section, we list detailed prompts used in our experiments.

## C.1 Database Styles

### C.1.1 Table(Column)

# stadium(Stadium_ID, Location, Name, Capacity, Highest, Lowest, Average)
# singer(Singer_ID, Name, Country, Song_Name, Song_release_year, Age, Is_male)
# concert(concert_ID, concert_Name, Theme, Stadium_ID, Year)
# singer_in_concert(concert_ID, Singer_ID)

### C.1.2 Table(Column)(PF)

# stadium(Stadium_ID, Location, Name, Capacity, Highest, Lowest, Average)
# singer(Singer_ID, Name, Country, Song_Name, Song_release_year, Age, Is_male)
# concert(concert_ID, concert_Name, Theme, Stadium_ID, Year)
# singer_in_concert(concert_ID, Singer_ID)
# primary keys = [stadium.Stadium_ID, singer.Singer_ID, concert.concert_ID, singer_in_concert.concert_ID]
# foreign keys = [concert.Stadium_ID = stadium.Stadium_ID, singer_in_concert.Singer_ID = singer.Singer_ID, singer_in_concert.concert_ID = concert.concert_ID]

### C.1.3 Create(NoPF)

```
create table stadium (
      Stadium_ID number,
      Location text,
      Name text,
      Capacity number,
      Highest number,
      Lowest number,
      Average number
)
create table singer (
      Singer_ID number,
      Name text,
      Country text,
      Song_Name text,
      Song_release_year text,
      Age number,
      Is_male others
)
create table concert (
      concert_ID number,
      concert_Name text,
      Theme text,
      Stadium_ID text,
      Year text
)
create table singer_in_concert (
      concert_ID number,
      Singer_ID text
)
```

### C.1.4 Create(EoC)

```
create table stadium (
      Stadium_ID number primary key,
      Location text,
```

```
        Name text,
        Capacity number,
        Highest number,
        Lowest number,
        Average number
)
create table singer (
        Singer_ID number primary key,
        Name text,
        Country text,
        Song_Name text,
        Song_release_year text,
        Age number,
        Is_male others
)
create table concert (
        concert_ID number primary key,
        concert_Name text,
        Theme text,
        Stadium_ID text references stadium(Stadium_ID),
        Year text
)
create table singer_in_concert (
        concert_ID number primary key references concert(concert_ID),
        Singer_ID text references singer(Singer_ID)
)
```

### C.1.5  Create(EoT)

```
create table stadium (
        Stadium_ID number,
        Location text,
        Name text,
        Capacity number,
        Highest number,
        Lowest number,
        Average number,
        primary key (Stadium_ID)
)
create table singer (
        Singer_ID number,
        Name text,
        Country text,
        Song_Name text,
        Song_release_year text,
        Age number,
        Is_male others,
        primary key (Singer_ID)
)
create table concert (
        concert_ID number,
        concert_Name text,
        Theme text,
```

```
        Stadium_ID text,
        Year text,
        primary key (concert_ID),
        foreign key (Stadium_ID) references stadium(Stadium_ID)
)
create table singer_in_concert (
        concert_ID number,
        Singer_ID text,
        primary key (concert_ID),
        foreign key (Singer_ID) references singer(Singer_ID),
        foreign key (concert_ID) references concert(concert_ID)
)
```

## C.2   Database Contents

We only use the `Table(Column)` and the `Create(EoT)` database styles in the following prompt examples. The other three database styles are similar. The rows of database contents is set to 3 in the following prompt examples.

### C.2.1   Table(Column)

\# stadium(Stadium_ID, Location, Name, Capacity, Highest, Lowest, Average)
/*
3 example rows from table stadium:

| Stadium_ID | Location | Name | Capacity | Highest | Lowest | Average |
|---|---|---|---|---|---|---|
| 1 | Raith Rovers | Stark's Park | 10104 | 4812 | 1294 | 2106 |
| 2 | Ayr United | Somerset Park | 11998 | 2363 | 1057 | 1477 |
| 3 | East Fife | Bayview Stadium | 2000 | 1980 | 533 | 864 |

**/
\# singer(Singer_ID, Name, Country, Song_Name, Song_release_year, Age, Is_male)
/*
3 example rows from table singer:

| Singer_ID | Name | Country | Song_Name | Song_release_year | Age | Is_male |
|---|---|---|---|---|---|---|
| 1 | Joe Sharp | Netherlands | You | 1992 | 52 | F |
| 2 | Timbaland | United States | Dangerous | 2008 | 32 | T |
| 3 | Justin Brown | France | Hey Oh | 2013 | 29 | T |

**/
\# concert(concert_ID, concert_Name, Theme, Stadium_ID, Year)
/*
3 example rows from table concert:

| concert_ID | concert_Name | Theme | Stadium_ID | Year |
|---|---|---|---|---|
| 1 | Auditions | Free choice | 1 | 2014 |
| 2 | Super bootcamp | Free choice 2 | 2 | 2014 |
| 3 | Home Visits | Bleeding Love | 2 | 2015 |

**/
\# singer_in_concert(concert_ID, Singer_ID) /*
3 example rows from table singer_in_concert:

| concert_ID | Singer_ID |
|---|---|
| 1 | 2 |
| 1 | 3 |
| 1 | 5 |

**/

### C.2.2 Create(EoT)

```
create table stadium (
      Stadium_ID number,
      Location text,
      Name text,
      Capacity number,
      Highest number,
      Lowest number,
      Average number,
      primary key (Stadium_ID)
)
/*
```
3 example rows from table stadium:

| Stadium_ID | Location | Name | Capacity | Highest | Lowest | Average |
|---|---|---|---|---|---|---|
| 1 | Raith Rovers | Stark's Park | 10104 | 4812 | 1294 | 2106 |
| 2 | Ayr United | Somerset Park | 11998 | 2363 | 1057 | 1477 |
| 3 | East Fife | Bayview Stadium | 2000 | 1980 | 533 | 864 |

```
**/
create table singer (
      Singer_ID number,
      Name text,
      Country text,
      Song_Name text,
      Song_release_year text,
      Age number,
      Is_male others,
      primary key (Singer_ID)
)
/*
```
3 example rows from table singer:

| Singer_ID | Name | Country | Song_Name | Song_release_year | Age | Is_male |
|---|---|---|---|---|---|---|
| 1 | Joe Sharp | Netherlands | You | 1992 | 52 | F |
| 2 | Timbaland | United States | Dangerous | 2008 | 32 | T |
| 3 | Justin Brown | France | Hey Oh | 2013 | 29 | T |

```
**/
create table concert (
      concert_ID number,
      concert_Name text,
      Theme text,
      Stadium_ID text,
      Year text,
      primary key (concert_ID),
      foreign key (Stadium_ID) references stadium(Stadium_ID)
)
/*
```
3 example rows from table concert:

| concert_ID | concert_Name | Theme | Stadium_ID | Year |
|---|---|---|---|---|
| 1 | Auditions | Free choice | 1 | 2014 |
| 2 | Super bootcamp | Free choice 2 | 2 | 2014 |
| 3 | Home Visits | Bleeding Love | 2 | 2015 |

```
**/
```

```
create table singer_in_concert (
      concert_ID number,
      Singer_ID text,
      primary key (concert_ID),
      foreign key (Singer_ID) references singer(Singer_ID),
      foreign key (concert_ID) references concert(concert_ID)
)
/*
3 example rows from table singer_in_concert:
concert_ID      Singer_ID
1      2
1      3
1      5
**/
```

### C.3    In-context Learning Prompts

We only use the Create(EoT) database styles in the following prompt examples. The other four database styles are similar. The rows of database contents is set to 3 in the following prompt examples. Under the few-shot setting, the first two shots are static exemplars and the last two shots are dynamic exemplars.

#### C.3.1    Zero-shot

role: system
content:
Given the database schema, you need to translate the question into the SQL query.

role: user
content:
Database schema:
```
create table stadium (
      Stadium_ID number,
      Location text,
      Name text,
      Capacity number,
      Highest number,
      Lowest number,
      Average number,
      primary key (Stadium_ID)
)
/*
3 example rows from table stadium:
```

| Stadium_ID | Location | Name | Capacity | Highest | Lowest | Average |
|---|---|---|---|---|---|---|
| 1 | Raith Rovers | Stark's Park | 10104 | 4812 | 1294 | 2106 |
| 2 | Ayr United | Somerset Park | 11998 | 2363 | 1057 | 1477 |
| 3 | East Fife | Bayview Stadium | 2000 | 1980 | 533 | 864 |

```
**/
create table singer (
      Singer_ID number,
      Name text,
      Country text,
      Song_Name text,
      Song_release_year text,
      Age number,
```

```
        Is_male others,
        primary key (Singer_ID)
)
/*
3 example rows from table singer:
Singer_ID       Name        Country       Song_Name       Song_release_year       Age       Is_male
1       Joe Sharp       Netherlands       You       1992       52       F
2       Timbaland       United States       Dangerous       2008       32       T
3       Justin Brown       France       Hey Oh       2013       29       T
**/
create table concert (
        concert_ID number,
        concert_Name text,
        Theme text,
        Stadium_ID text,
        Year text,
        primary key (concert_ID),
        foreign key (Stadium_ID) references stadium(Stadium_ID)
)
/*
3 example rows from table concert:
concert_ID       concert_Name       Theme       Stadium_ID       Year
1       Auditions       Free choice       1       2014
2       Super bootcamp       Free choice 2       2       2014
3       Home Visits       Bleeding Love       2       2015
**/
create table singer_in_concert (
        concert_ID number,
        Singer_ID text,
        primary key (concert_ID),
        foreign key (Singer_ID) references singer(Singer_ID),
        foreign key (concert_ID) references concert(concert_ID)
)
/*
3 example rows from table singer_in_concert:
concert_ID       Singer_ID
1       2
1       3
1       5
**/
Question: How many singers do we have?
```

### C.3.2   Few-shot

role: system
content:
Given the database schema, you need to translate the question into the SQL query.

role: user
content:
Database schema:
```
create table ACCOUNTS (
        custid number,
```

```
        name text,
        primary key (custid)
)
/*
3 example rows from table ACCOUNTS:
custid      name
1       Brown
2       Wang
3       O'mahony
**/
create table SAVINGS (
        custid number,
        balance number,
        primary key (custid),
        foreign key (custid) references ACCOUNTS(custid)
)
/*
3 example rows from table SAVINGS:
custid      balance
1       200000.0
2       999999999.0
3       230000.0
**/
create table CHECKING (
        custid number,
        balance number,
        primary key (custid),
        foreign key (custid) references ACCOUNTS(custid)
)
/*
3 example rows from table CHECKING:
custid      balance
1       10000.0
2       2000.0
3       3000.0
**/
```

Question: Find the name and savings balance of the top 3 accounts with the highest saving balance sorted by savings balance in descending order.

role: assistant
content:
```
SELECT T1.name , T2.balance FROM accounts AS T1 JOIN savings AS T2 ON T1.custid = T2.custid ORDER BY T2.balance DESC LIMIT 3
```

role: user
content:
Database schema:
```
create table flight (
        flno number,
        origin text,
        destination text,
```

```
        distance number,
        departure_date time,
        arrival_date time,
        price number,
        aid number,
        primary key (flno),
        foreign key (aid) references aircraft(aid)
)
/*
3 example rows from table flight:
flno     origin      destination     distance     departure_date     arrival_date     price
    aid
99      Los Angeles     Washington D.C.     2308     04/12/2005 09:30     04/12/2005 09:40
    235.98     1
13      Los Angeles     Chicago     1749     04/12/2005 08:45     04/12/2005 08:45
    220.98     3
346     Los Angeles     Dallas     1251     04/12/2005 11:50     04/12/2005 07:05     182
    2
**/
create table aircraft (
        aid number,
        name text,
        distance number,
        primary key (aid)
)
/*
3 example rows from table aircraft:
aid     name     distance
1      Boeing 747-400     8430
2      Boeing 737-800     3383
3      Airbus A340-300     7120
**/
create table employee (
        eid number,
        name text,
        salary number,
        primary key (eid)
)
/*
3 example rows from table employee:
eid     name     salary
242518965     James Smith     120433
141582651     Mary Johnson     178345
11564812     John Williams     153972
**/
create table certificate (
        eid number,
        aid number,
        primary key (eid),
        foreign key (aid) references aircraft(aid),
        foreign key (eid) references employee(eid)
```

)
/*
3 example rows from table certificate:
eid         aid
11564812        2
11564812        10
90873519        6
**/
Question: Which destination has least number of flights?

role: assistant
content:
SELECT destination FROM Flight GROUP BY destination ORDER BY count(*) LIMIT 1

role: user
content:
Database schema:
create table Activity (
        actid number,
        activity_name text,
        primary key (actid)
)
/*
3 example rows from table Activity:
actid       activity_name
770         Mountain Climbing
771         Canoeing
772         Kayaking
**/
create table Participates_in (
        stuid number,
        actid number,
        foreign key (actid) references Activity(actid),
        foreign key (stuid) references Student(StuID)
)
/*
3 example rows from table Participates_in:
stuid       actid
1001        770
1001        771
1001        777
**/
create table Faculty_Participates_in (
        FacID number,
        actid number,
        foreign key (actid) references Activity(actid),
        foreign key (FacID) references Faculty(FacID)
)
/*
3 example rows from table Faculty_Participates_in:
FacID           actid

```
1082      784
1082      785
1082      790
**/
create table Student (
      StuID number,
      LName text,
      Fname text,
      Age number,
      Sex text,
      Major number,
      Advisor number,
      city_code text,
      primary key (StuID)
)
/*
3 example rows from table Student:
StuID      LName      Fname      Age      Sex      Major      Advisor      city_code
1001      Smith      Linda      18      F      600      1121      BAL
1002      Kim      Tracy      19      F      600      7712      HKG
1003      Jones      Shiela      21      F      600      7792      WAS
**/
create table Faculty (
      FacID number,
      Lname text,
      Fname text,
      Rank text,
      Sex text,
      Phone number,
      Room text,
      Building text,
      primary key (FacID)
)
/*
3 example rows from table Faculty:
FacID      Lname      Fname      Rank      Sex      Phone      Room      Building
1082      Giuliano      Mark      Instructor      M      2424      224      NEB
1121      Goodrich      Michael      Professor      M      3593      219      NEB
1148      Masson      Gerald      Professor      M      3402      224B      NEB
**/
Question: How many female Professors do we have?

role: assistant
content:
SELECT count(*) FROM Faculty WHERE Sex = 'F' AND Rank = "Professor"

role: user
content:
Database schema:
create table region (
      Region_ID number,
```

```
        Region_name text,
        Date text,
        Label text,
        Format text,
        Catalogue text,
        primary key (Region_ID)
)
/*
3 example rows from table region:
Region_ID       Region_name       Date       Label       Format       Catalogue
1       United Kingdom       1 July 2002       Parlophone       CD       540 3622
2       United Kingdom       1 July 2002       Parlophone       2× LP       539 9821
3       Japan       3 July 2002       Toshiba-EMI       CD       TOCP-66045
**/
create table party (
        Party_ID number,
        Minister text,
        Took_office text,
        Left_office text,
        Region_ID number,
        Party_name text,
        primary key (Party_ID),
        foreign key (Region_ID) references region(Region_ID)
)
/*
3 example rows from table party:
Party_ID       Minister       Took_office       Left_office       Region_ID       Party_name
1       Dr. Kwame Nkrumah (MP)       1957       1958       1       Convention Peoples Party
2       Kojo Botsio (MP)       1958       1959       2       Progress Party
3       Ebenezer Ako-Adjei (MP)       1959       1960       3       3
**/
create table member (
        Member_ID number,
        Member_Name text,
        Party_ID text,
        In_office text,
        primary key (Member_ID),
        foreign key (Party_ID) references party(Party_ID)
)
/*
3 example rows from table member:
Member_ID       Member_Name       Party_ID       In_office
1       Hon Tony Abbott       3       1994–present
2       Hon Dick Adams       2       1993–2013
3       Anthony Albanese       2       1996–present
**/
create table party_events (
        Event_ID number,
        Event_Name text,
        Party_ID number,
        Member_in_charge_ID number,
```

primary key (Event_ID),
        foreign key (Member_in_charge_ID) references member(Member_ID),
        foreign key (Party_ID) references party(Party_ID)
)
/*
3 example rows from table party_events:

| Event_ID | Event_Name | Party_ID | Member_in_charge_ID |
| --- | --- | --- | --- |
| 1 | Annaual Meeting | 1 | 4 |
| 2 | Conference | 1 | 12 |
| 3 | Annaual Meeting | 2 | 2 |

**/
Question: How many parties do we have?

role: assistant
content:
SELECT count(DISTINCT party_name) FROM party

role: user
content:
Database schema:
create table stadium (
        Stadium_ID number,
        Location text,
        Name text,
        Capacity number,
        Highest number,
        Lowest number,
        Average number,
        primary key (Stadium_ID)
)
/*
3 example rows from table stadium:

| Stadium_ID | Location | Name | Capacity | Highest | Lowest | Average |
| --- | --- | --- | --- | --- | --- | --- |
| 1 | Raith Rovers | Stark's Park | 10104 | 4812 | 1294 | 2106 |
| 2 | Ayr United | Somerset Park | 11998 | 2363 | 1057 | 1477 |
| 3 | East Fife | Bayview Stadium | 2000 | 1980 | 533 | 864 |

**/
create table singer (
        Singer_ID number,
        Name text,
        Country text,
        Song_Name text,
        Song_release_year text,
        Age number,
        Is_male others,
        primary key (Singer_ID)
)
/*
3 example rows from table singer:

| Singer_ID | Name | Country | Song_Name | Song_release_year | Age | Is_male |
| --- | --- | --- | --- | --- | --- | --- |
| 1 | Joe Sharp | Netherlands | You | 1992 | 52 | F |

| 2 | Timbaland | United States | Dangerous | 2008 | 32 | T |
| 3 | Justin Brown | France | Hey Oh | 2013 | 29 | T |

**\*/

```
create table concert (
    concert_ID number,
    concert_Name text,
    Theme text,
    Stadium_ID text,
    Year text,
    primary key (concert_ID),
    foreign key (Stadium_ID) references stadium(Stadium_ID)
)
```

/*

3 example rows from table concert:

| concert_ID | concert_Name | Theme | Stadium_ID | Year |
| 1 | Auditions | Free choice | 1 | 2014 |
| 2 | Super bootcamp | Free choice 2 | 2 | 2014 |
| 3 | Home Visits | Bleeding Love | 2 | 2015 |

**\*/

```
create table singer_in_concert (
    concert_ID number,
    Singer_ID text,
    primary key (concert_ID),
    foreign key (Singer_ID) references singer(Singer_ID),
    foreign key (concert_ID) references concert(concert_ID)
)
```

/*

3 example rows from table singer_in_concert:

| concert_ID | Singer_ID |
| 1 | 2 |
| 1 | 3 |
| 1 | 5 |

**\*/

Question: How many singers do we have?

### C.3.3 Auto-CoT

role: system
content:
Given the database schema, you need to translate the question into the SQL query.

role: user
content:
Database schema:

```
create table ACCOUNTS (
    custid number,
    name text,
    primary key (custid)
)
```

/*

3 example rows from table ACCOUNTS:

| custid | name |
| 1 | Brown |

2      Wang
3      O'mahony
**/
create table SAVINGS (
     custid number,
     balance number,
     primary key (custid),
     foreign key (custid) references ACCOUNTS(custid)
)
/*
3 example rows from table SAVINGS:
custid      balance
1      200000.0
2      999999999.0
3      230000.0
**/
create table CHECKING (
     custid number,
     balance number,
     primary key (custid),
     foreign key (custid) references ACCOUNTS(custid)
)
/*
3 example rows from table CHECKING:
custid      balance
1      10000.0
2      2000.0
3      3000.0
**/
Question: Find the name and savings balance of the top 3 accounts with the highest saving balance sorted by savings balance in descending order.

role: assistant
content:
Let's think step by step.
According to "savings balance", columns [SAVINGS.balance] may be used.
According to "accounts", columns [ACCOUNTS.name] may be used.
Values [3] may be used.
So the final answer is:
SELECT T1.name , T2.balance FROM accounts AS T1 JOIN savings AS T2 ON T1.custid = T2.custid ORDER BY T2.balance DESC LIMIT 3

role: user
content:
Database schema:
create table flight (
     flno number,
     origin text,
     destination text,
     distance number,
     departure_date time,

```
        arrival_date time,
        price number,
        aid number,
        primary key (flno),
        foreign key (aid) references aircraft(aid)
)
/*
3 example rows from table flight:
flno      origin      destination     distance     departure_date      arrival_date      price
    aid
99      Los Angeles     Washington D.C.     2308      04/12/2005 09:30      04/12/2005 09:40
    235.98      1
13      Los Angeles     Chicago      1749      04/12/2005 08:45      04/12/2005 08:45
    220.98      3
346      Los Angeles     Dallas      1251      04/12/2005 11:50      04/12/2005 07:05      182
    2
**/
create table aircraft (
        aid number,
        name text,
        distance number,
        primary key (aid)
)
/*
3 example rows from table aircraft:
aid      name      distance
1      Boeing 747-400      8430
2      Boeing 737-800      3383
3      Airbus A340-300      7120
**/
create table employee (
        eid number,
        name text,
        salary number,
        primary key (eid)
)
/*
3 example rows from table employee:
eid      name      salary
242518965      James Smith      120433
141582651      Mary Johnson      178345
11564812      John Williams      153972
**/
create table certificate (
        eid number,
        aid number,
        primary key (eid),
        foreign key (aid) references aircraft(aid),
        foreign key (eid) references employee(eid)
)
/*
```

3 example rows from table certificate:

| eid | aid |
|-----|-----|
| 11564812 | 2 |
| 11564812 | 10 |
| 90873519 | 6 |

**/
Question: Which destination has least number of flights?

role: assistant
content:
Let's think step by step.
According to "flights", columns [flight.destination] may be used.
Values [1] may be used.
So the final answer is:
SELECT destination FROM Flight GROUP BY destination ORDER BY count(*) LIMIT 1

role: user
content:
Database schema:
create table Student (
        StuID number,
        LName text,
        Fname text,
        Age number,
        Sex text,
        Major number,
        Advisor number,
        city_code text,
        primary key (StuID)
)
/*
3 example rows from table Student:

| StuID | LName | Fname | Age | Sex | Major | Advisor | city_code |
|-------|-------|-------|-----|-----|-------|---------|-----------|
| 1001 | Smith | Linda | 18 | F | 600 | 1121 | BAL |
| 1002 | Kim | Tracy | 19 | F | 600 | 7712 | HKG |
| 1003 | Jones | Shiela | 21 | F | 600 | 7792 | WAS |

**/
create table Dorm (
        dormid number,
        dorm_name text,
        student_capacity number,
        gender text
)
/*
3 example rows from table Dorm:

| dormid | dorm_name | student_capacity | gender |
|--------|-----------|------------------|--------|
| 100 | Smith Hall | 85 | X |
| 110 | Bud Jones Hall | 116 | M |
| 140 | Fawlty Towers | 355 | X |

**/
create table Dorm_amenity (

```
        amenid number,
        amenity_name text
)
/*
3 example rows from table Dorm_amenity:
amenid        amenity_name
900        TV Lounge
901        Study Room
902        Pub in Basement
**/
create table Has_amenity (
        dormid number,
        amenid number,
        foreign key (amenid) references Dorm_amenity(amenid),
        foreign key (dormid) references Dorm(dormid)
)
/*
3 example rows from table Has_amenity:
dormid        amenid
109        900
109        901
109        903
**/
create table Lives_in (
        stuid number,
        dormid number,
        room_number number,
        foreign key (dormid) references Dorm(dormid),
        foreign key (stuid) references Student(StuID)
)
/*
3 example rows from table Lives_in:
stuid        dormid        room_number
1001        109        105
1002        100        112
1003        100        124
**/
Question: How many dorms have amenities?
```

role: assistant
content:
Let's think step by step.
According to "dorms have amenities ?", columns [Has_amenity.dormid] may be used.
So the final answer is:
SELECT count(DISTINCT dormid) FROM has_amenity

role: user
content:
Database schema:
create table Student (
        StuID number,

```
        LName text,
        Fname text,
        Age number,
        Sex text,
        Major number,
        Advisor number,
        city_code text,
        primary key (StuID)
)
/*
3 example rows from table Student:
StuID    LName      Fname      Age     Sex     Major     Advisor     city_code
1001     Smith      Linda      18      F       600       1121        BAL
1002     Kim        Tracy      19      F       600       7712        HKG
1003     Jones      Shiela     21      F       600       7792        WAS
**/
create table Dorm (
        dormid number,
        dorm_name text,
        student_capacity number,
        gender text
)
/*
3 example rows from table Dorm:
dormid      dorm_name       student_capacity        gender
100         Smith Hall      85      X
110         Bud Jones Hall      116      M
140         Fawlty Towers       355      X
**/
create table Dorm_amenity (
        amenid number,
        amenity_name text
)
/*
3 example rows from table Dorm_amenity:
amenid       amenity_name
900          TV Lounge
901          Study Room
902          Pub in Basement
**/
create table Has_amenity (
        dormid number,
        amenid number,
        foreign key (amenid) references Dorm_amenity(amenid),
        foreign key (dormid) references Dorm(dormid)
)
/*
3 example rows from table Has_amenity:
dormid        amenid
109        900
109        901
```

109      903
**/
create table Lives_in (
       stuid number,
       dormid number,
       room_number number,
       foreign key (dormid) references Dorm(dormid),
       foreign key (stuid) references Student(StuID)
)
/*
3 example rows from table Lives_in:
stuid       dormid      room_number
1001      109      105
1002      100      112
1003      100      124
**/
Question: How many diffrent dorm amenities are there?

role: assistant
content:
Let's think step by step.
According to "dorm amenities are there ?", tables [Dorm_amenity] may be used.
So the final answer is:
SELECT count(*) FROM dorm_amenity

role: user
content:
Database schema:
create table stadium (
       Stadium_ID number,
       Location text,
       Name text,
       Capacity number,
       Highest number,
       Lowest number,
       Average number,
       primary key (Stadium_ID)
)
/*
3 example rows from table stadium:

| Stadium_ID | Location | Name | Capacity | Highest | Lowest | Average |
|---|---|---|---|---|---|---|
| 1 | Raith Rovers | Stark's Park | 10104 | 4812 | 1294 | 2106 |
| 2 | Ayr United | Somerset Park | 11998 | 2363 | 1057 | 1477 |
| 3 | East Fife | Bayview Stadium | 2000 | 1980 | 533 | 864 |

**/
create table singer (
       Singer_ID number,
       Name text,
       Country text,
       Song_Name text,
       Song_release_year text,

```
    Age number,
    Is_male others,
    primary key (Singer_ID)
)
/*
3 example rows from table singer:
Singer_ID    Name         Country        Song_Name      Song_release_year    Age      Is_male
1     Joe Sharp       Netherlands     You      1992     52      F
2     Timbaland       United States    Dangerous    2008     32       T
3     Justin Brown      France     Hey Oh     2013      29      T
**/
create table concert (
    concert_ID number,
    concert_Name text,
    Theme text,
    Stadium_ID text,
    Year text,
    primary key (concert_ID),
    foreign key (Stadium_ID) references stadium(Stadium_ID)
)
/*
3 example rows from table concert:
concert_ID    concert_Name      Theme      Stadium_ID      Year
1     Auditions     Free choice     1     2014
2     Super bootcamp      Free choice 2     2      2014
3     Home Visits      Bleeding Love     2      2015
**/
create table singer_in_concert (
    concert_ID number,
    Singer_ID text,
    primary key (concert_ID),
    foreign key (Singer_ID) references singer(Singer_ID),
    foreign key (concert_ID) references concert(concert_ID)
)
/*
3 example rows from table singer_in_concert:
concert_ID    Singer_ID
1     2
1     3
1     5
**/
Question: How many singers do we have?
```