# OpenReview forum: "ACT-SQL: In-Context Learning for Text-to-SQL with Automatically-Generated Chain-of-Thought"
_EMNLP/2023/Conference — EMNLP 2023 Findings_

### Official Review · Reviewer_tJPB · 2023-07-28

**Typos Grammar Style And Presentation Improvements:** N/A
**Soundness:** 2

**Excitement:**

3: Ambivalent: It has merits (e.g., it reports state-of-the-art results, the idea is nice), but there are key weaknesses (e.g., it describes incremental work), and it can significantly benefit from another round of revision. However, I won't object to accepting it if my co-reviewers champion it.

**Missing References:**

N/A

**Paper Topic And Main Contributions:**

The paper proposes to automatically generate the chain-of-of-thought reasoning steps for the text-to-SQL (ACT-SQL). The authors suggest using hybrid demonstration examples, which contains n_s static examples for all test case, and n_d dynamic examples selected based on specific test cases. Additionally, these demonstration examples are compounded with automatically generated chain-of-thought reasoning steps.
The paper compares different database prompt styles in the zero-shot and few-shot text-to-SQL.
The results demonstrate that automatically generated chain-of-thought steps benefit LLMs on text-to-SQL generation.

**Questions For The Authors:**

1. See above for the statistical significance test. What's the p-value in the null hypothesis when comparing Few-shot and ACT-SQL in EM/EX/TS?
2. The authors use a neural model + rule-based method to create chain-of-thought reasoning steps for demonstrations. How does it compare to the chain-of-thought reasoning steps obtained by using "let's think step by step" to prompt ChatGPT. [1]

[1] Kojima, Takeshi, et al. "Large language models are zero-shot reasoners." Advances in neural information processing systems 35 (2022): 22199-22213.

**Reasons To Accept:**

1. The author presents a comprehensive comparison of various database prompt styles in the zero-shot as well as a comparison between using trivial few-shot examples and using ACT few-shot examples with different database prompt styles.
2. The experiments cover multiple single-turn and multi-turn text-to-SQL datasets, where the proposed ACT-SQL outperforms the baseline Few-shot method in a majority of these datasets.
3. The paper is well-written and easy to follow.

**Reasons To Reject:**

1. My biggest concern is the effectiveness of the proposed ACT-SQL. In Table 10, the author presents a comparison between using trivial few-shot examples and using ACT few-shot examples with different database prompt styles. It seems the difference between Few-shot and ACT-SQL is very small (<1% for most cases) for Table(Column), Table(Column) PF, Create(NoPF), and relatively large (about 2%) for Create(EoC) and Create(EoT). Given that the Spider dev set only contains about 1K examples, it seems important to do a statistical significance test when comparing Few-shot and ACT-SQL. The same statistical significance test should be applied to Table 4,5,6,7,8 as the results of Few-shot and ACT-SQL are very close (ACT-SQL even underperforms Few-shot in Table 6,7,8 for TS accuracy).
I believe addressing this concern during the rebuttal would be essential.

**Reproducibility:**

4: Could mostly reproduce the results, but there may be some variation because of sample variance or minor variations in their interpretation of the protocol or method.

**Reviewer Confidence:**

5: Positive that my evaluation is correct. I read the paper very carefully and I am very familiar with related work.

---

> ### Author Rebuttal · Authors · 2023-08-29
>
> Thanks for your valuable suggestions.
>
> **Q1:** Statistical significance test between Few-shot and ACT-SQL?
>
> **A1:** We complete some more experiments with several different random seeds and then compute the p-value. Under the Spider dev set, the p-value in EM/EX/TS is 0.004/0.048/0.016. These p-values are lower than 0.05.
>
> **Q2:** How about using "let's think step by step" to prompt GPT?
>
> **A2:**
>
> **a)** Actually according to Figure 1 and Figure 2 in our paper, we have added "let's think step by step" into the prompt.
>
> **b)** If we only use "let's think step by step" to prompt GPT without any exemplar, the performance would definitely decrease, since exemplars play a very important role for the in-context learning. We attempt to do the experiment with GPT-3.5-turbo. The TS score is only 59.4, which is lower than ACT-SQL TS score 71.4.

---

### Official Review · Reviewer_sUqE · 2023-08-05

**Soundness:** 3

**Excitement:**

3: Ambivalent: It has merits (e.g., it reports state-of-the-art results, the idea is nice), but there are key weaknesses (e.g., it describes incremental work), and it can significantly benefit from another round of revision. However, I won't object to accepting it if my co-reviewers champion it.

**Paper Topic And Main Contributions:**

The paper focuses on the challenge of prompt designing in the text-to-SQL task, aiming to enhance LLMs' reasoning capabilities when generating SQL queries. The authors introduce a method called ACT-SQL to automatically generate auto-CoT exemplars without manual labeling, making the process cost-saving. The approach is extended to multi-turn text-to-SQL tasks, and experimental results demonstrate that ACT-SQL achieves state-of-the-art performance on the Spider dev set among existing in-context learning approaches.

**Reasons To Accept:**

1. This paper introduces chain-of-thought (CoT) prompt design, inspired by schema linking, represents a fresh perspective in the text-to-SQL task.
2. The ACT-SQL method for automatically generating auto-CoT exemplars without manual labeling streamlines the process and enhances efficiency.
3. The application of the approach to multi-turn text-to-SQL tasks demonstrates its versatility and potential adaptability to more complex dialog scenarios.
4. By attaining SOTA performance on the Spider dev set among existing in-context learning approaches, the paper showcases the effectiveness of the proposed method.

**Reasons To Reject:**

1. While the paper introduces the chain-of-thought (CoT) prompt design and in-context learning techniques, it does not appear to be the first work to utilize these concepts in the text-to-SQL task. Many similar works in the public domain have explored these ideas, leading to difficulty in distinguishing the novelty of this particular research.
2. The paper seems to lack a comprehensive comparison with related similar works that have used CoT or in-context learning techniques. A detailed analysis of how the proposed method differs from or improves upon existing approaches would enhance the paper's credibility and relevance.
3. While the paper claims state-of-the-art performance on the Spider dev set, the extent or significance of this performance improvement is not made clear. A more transparent presentation of the performance metrics and comparison with benchmarks would provide a robust validation of the claimed achievements.
4. Without clear differentiation from existing works and a detailed comparison with similar techniques, the paper's contributions may be perceived as incremental or lacking substantial novelty. This could affect the reception and perceived value of the work within the NLP community.

**Reproducibility:**

4: Could mostly reproduce the results, but there may be some variation because of sample variance or minor variations in their interpretation of the protocol or method.

**Reviewer Confidence:**

4: Quite sure. I tried to check the important points carefully. It's unlikely, though conceivable, that I missed something that should affect my ratings.

---

> ### Author Rebuttal · Authors · 2023-08-29
>
> Thanks for your valuable suggestions.
>
> **Q1:** What is the novelty of this research?
>
> **A1:**
>
> **a)** The previous work DIN-SQL [1] has applied the chain-of-thought (CoT) method for text-to-SQL. However, DIN-SQL needs manually labeled exemplars. Manually labeled CoT has some disadvantages. First, it is cost-consuming and time-consuming. Second, it needs domain experts to design and write the CoT. Third, it is unfriendly for exemplar selection strategy, since dynamic exemplars lack CoTs unless the whole train set is labeled by human. We propose the method to automatically generate the CoT for the given exemplar, which is cost-saving and time-saving.
>
> **b)** When handling one testing case, ACT-SQL calls LLMs' API only once while DIN-SQL calls LLMs' API 4 times. Our approach is faster and cheaper. In real-world scenarios and applications, users are always very sensitive to their user experience. Therefore, optimizing the response time is an important novelty of our research. Our approach decreases the total response time for handling one user question. Thus real-world users would not be impatient when waiting for the response.
>
> **c)** Under the same LLM, we achieve the SOTA performance on Spider dev set with the in-context learning method. The TS score is 71.4 when using GPT-3.5-turbo. The TS score is 74.5 when using GPT-4. Furthermore, to the best of our knowledge, we are also the first to apply the in-context learning method on the multi-turn text-to-SQL task.
>
> **Q2:** Comparison with similar works?
>
> **A2:** Actually Table 3 in our paper has shown the comparison with related similar works. The following table shows the detailed comparison. Comparing with [2] and [3], ACT-SQL achieves the much better performance on Spider dev set since the CoT method is introduced. Comparing with DIN-SQL [1], ACT-SQL performs a bit better than DIN-SQL and uses less LLMs' API call, i.e. our approach is more effective and efficient. Furthermore, our approach can automatically generate the CoT for the given exemplar, which avoids the cost-consuming human labeling by domain experts.
>
> |           Approach            | Spider Dev EX/TS | # API Calls | Automatically | CoT  |
> | :---------------------------: | :--------------: | :---------: | :-----------: | :--: |
> |      Rajkumar et al. [2]      |    67.0/55.1     |      1      |      yes      |  no  |
> | Chang and Fosler-Lussier. [3] |      76.8/-      |      1      |      yes      |  no  |
> |          DIN-SQL [1]          |    82.8/74.2     |      4      |      no       | yes  |
> |        ACT-SQL (Ours)         |    82.9/74.5     |      1      |      yes      | yes  |
>
> **Q3:** The extent or significance of the performance improvement is not made clear.
>
> **A3:**
>
> **a)** Comparing with previous works, we achieve a better performance with less LLMs' API call. This indicates that we propose a approach to prompt GPT for the text-to-SQL task in a better, faster and cheaper way.
>
> **b)** We also complete some more experiments with several different random seeds and do the statistical significance test to reveal the performance improvement from the ordinary few-shot method to our ACT-SQL. The p-value in the null hypothesis when comparing few-shot and ACT-SQL in EM/EX/TS is 0.004/0.048/0.016. These p-values are lower than 0.05. Thus our approach is effective.
>
> **Q4:** No clear differentiation from existing works.
>
> **A4:** The novelty of our research and the comparison with similar works are stated in A1 and A2.
>
> [1] Pourreza and Rafiei. 2023. DIN-SQL: Decomposed In-Context Learning of Text-to-SQL with Self-Correction.
>
> [2] Rajkumar et al. 2022. Evaluating the Text-to-SQL Capabilities of Large Language Models.
>
> [3] Chang and Fosler-Lussier. 2023. How to Prompt LLMs for Text-to-SQL: A Study in Zero-shot, Single-domain, and Cross-domain Settings.

---

### Official Review · Reviewer_5VBy · 2023-08-07

**Typos Grammar Style And Presentation Improvements:** Line 439 lower case `We`.
**Soundness:** 4

**Excitement:**

3: Ambivalent: It has merits (e.g., it reports state-of-the-art results, the idea is nice), but there are key weaknesses (e.g., it describes incremental work), and it can significantly benefit from another round of revision. However, I won't object to accepting it if my co-reviewers champion it.

**Paper Topic And Main Contributions:**

In this work, the authors proposed ACT-SQL that generates the Chain-of-Thought automatically for the training set. These examples are beneficial for the in-context learning. Only one LLM API call is required for this method compared to DIN-SQL’s 4 API calls. This method can be extended to multi-turn Text-to-SQL problems. To mitigate multi-turn text-to-SQL, the LLM is leveraged to translate the sequence of utterances into a single question to facilitate schema linking. The experiments have shown that the proposed method achieved state-of-the-art performance on Spider dataset under the in-context learning category.


**Questions For The Authors:**

1. Are table linking from line 325 to line 327 robust? Considering that some of the tables in JOIN part only serve as bridge function (connect two tables with foreign keys), the question may not explicitly mention those tables. In this case, enforcing the linking between tables and utterance chunks may not be reasonable.
2. I found that mixing randomly selected exemplars and test-case-specific exemplars yields the best performance. Any insights on this? Why didn’t more test-case-specific exemplars improve the performance?
Any evidence to show the claim from line 534-538 (in-context learning alleviate the performance gap issue between dev set and test set)
3. Comments: In line 570, the authors claim that the proposed in-context learning method is comparable to finetuned models. I think the authors refer to GAZP+BERT, because RASAT+PICARD is much higher than in-context learning. However, in terms of model size, GAZP+BERT is much smaller than GPT-3.5. The author should make careful claims by providing accurate assumptions.


**Reasons To Accept:**

1. Auto-generate Chain-of-Thought prompt is cost-saving.
2. By combining traditional query rewriting strategy, the proposed method can be easily extended to multi-turn text-to-SQL task.
3. Incremental contributions on studying prompting style and hybrid exemplar selection strategy.


**Reasons To Reject:**

1. In terms of schema linking based chain-of-thought prompt design, there is no significant difference from (Pourreza and Rafiei, 2023).
2. More on questions part.


**Reproducibility:**

4: Could mostly reproduce the results, but there may be some variation because of sample variance or minor variations in their interpretation of the protocol or method.

**Reviewer Confidence:**

4: Quite sure. I tried to check the important points carefully. It's unlikely, though conceivable, that I missed something that should affect my ratings.

---

> ### Author Rebuttal · Authors · 2023-08-29
>
> Thanks for your valuable suggestions.
>
> **Q1:** Are table linking robust?
>
> **A1:**
>
> **a)** In most cases, table names of bridge tables have semantics and can be mentioned in the question. Consider the following example. The bridge table "student_course" can still be mentioned in the question.
>
> **Question:** What courses does the student named Tom take?
>
> **SQL:** SELECT T3.course_name FROM student as T1 JOIN student_course as T2 JOIN course as T3 ON T1.student_id = T2.student_id AND T2.course_id = T3.course_id WHERE T1.student_name = "Tom"
>
> **b)** Since exemplars come from train set, their schema linking processes are completed with the rule-based method discussed in our paper. These exemplars are enforced to link tables with question chunks, which may lead to wrong schema linking. The following example shows the wrong table linking.
>
> **Database schema:**
>
> create table college (
>
> $\quad$id text,
>
> $\quad$name text,
>
> $\quad$primary key (id)
>
> )
>
> create table department (
>
> $\quad$id text,
>
> $\quad$name text,
>
> $\quad$cid text,
>
> $\quad$primary key (id),
>
> $\quad$foreign key (cid) references college(id)
>
> )
>
> create table student (
>
> $\quad$id text,
>
> $\quad$name text,
>
> $\quad$did text,
>
> $\quad$primary key (id),
>
> $\quad$foreign key (did) references department(did)
>
> )
>
> **Question:** List all student names from the college named XXX.
>
> **CoT:**
>
> Let's think step by step.
>
> According to "student names", columns [student.name] may be used.
>
> According to "college named", columns [college.name] may be used.
>
> According to "from", tables [department] may be used.
>
> Values [XXX] may be used.
>
> So the final answer is:
>
> SELECT T3.name FROM college as T1 JOIN department as T2 JOIN student as T3 ON T1.id = T2.cid AND T2.id = T3.did WHERE T1.name = "XXX"
>
> The linking between the bridge table "college" and the question word "from" is wrong. However, during the testing phase, LLMs would generate the CoT and complete the inference for the testing case. Although wrong table linking may appear in exemplars, LLMs like GPT have the ability to avoid unreasonable schema linking. Following is the example completed by GPT where we add the wrong exemplar mentioned above.
>
> **Database schema:**
>
> create table continent (
>
> $\quad$name text,
>
> $\quad$primary key (name)
>
> )
>
> create table country (
>
> $\quad$name text,
>
> $\quad$continent_name text,
>
> $\quad$primary key (name),
>
> $\quad$foreign key (continent_name) references continent(name)
>
> )
>
> create table singer (
>
> $\quad$id text,
>
> $\quad$name text,
>
> $\quad$country_name text,
>
> $\quad$foreign key (country_name) references country(name)
>
> )
>
> **Question:** What are singer names from Asia?
>
> **CoT:**
>
> Let's think step by step.
>
> According to "singer names", columns [singer.name] may be used.
>
> According to "from Asia", columns [continent.name] may be used.
>
> According to "singer names from", tables [singer] may be used.
>
> Values [Asia] may be used.
>
> So the final answer is:
>
> SELECT name FROM singer WHERE country_name IN (SELECT name FROM country WHERE continent_name = 'Asia')
>
> **Q2:** Why mixing exemplars but not more test-case-specific exemplars?
>
> **A2:**
>
> **a)** The previous work [1] has shown that exemplar selection strategy should balance diversity and similarity. The diversity would suffer if more test-case-specific exemplars are selected.
>
> **b)** Diversity is important for exemplar selection strategy. With different DB schema, similar questions can correspond to different SQL queries. Diverse exemplars can provide different SQL patterns.
>
> **Q3:** Does in-context learning alleviate the performance gap between dev set and test set?
>
> **A3:** Yes.
>
> **a)** The following table shows some previous works' performances on Spider dev set and test set. For finetuned approaches like RESDSQL-3B + NatSQL and Graphix-3B + PICARD, the performances both drop from dev set to test set. On the contrary, for in-context learning approaches like DIN-SQL and C3, the performances both increase from dev set to test set.
>
> **b)** After all, finetuned models are selected by the dev set performance, which would lead to the overfitting on dev set and the performance dropping on test set. For in-context learning approaches, dev set and test set are equal to the model. Performances between dev set and test set are only affected by the dataset feature.
>
> |      Previous Work      |     Method Type     | Spider Dev EX | Spider Test Ex |
> | :---------------------: | :-----------------: | :-----------: | :------------: |
> | RESDSQL-3B + NatSQL [2] | finetuned approach  |     84.1      |      79.9      |
> |  Graphix-3B+PICARD [3]  | finetuned approach  |     81.0      |      77.6      |
> |       DIN-SQL [4]       | in-context learning |     82.8      |      85.3      |
> |         C3 [5]          | in-context learning |     81.8      |      82.3      |
>
> **Q4:** ACT-SQL is comparable with GAZP+BERT but not RASAT+PICARD.
>
> **A4:** Sorry, we didn't make a clear statement in our paper. ACT-SQL is comparable with GAZP+BERT ($\approx$ 0.2B params) but not RASAT+PICARD ($\approx$ 3B params). Experiment results show that LLMs have the ability to handle the multi-turn text-to-SQL task. However, there is still a large room between our method and the SOTA model. Improving LLMs' performance on this difficult task is a challenging future work. We just complete the initial exploration.
>
> [1] Nan et al. 2023. Enhancing Few-shot Text-to-SQL Capabilities of Large Language Models: A Study on Prompt Design Strategies.
>
> [2] Li et al. 2023. RESDSQL: Decoupling Schema Linking and Skeleton Parsing for Text-to-SQL.
>
> [3] Li et al. 2023. Graphix-T5: Mixing Pre-Trained Transformers with Graph-Aware Layers for Text-to-SQL Parsing.
>
> [4] Pourreza and Rafiei. 2023. DIN-SQL: Decomposed In-Context Learning of Text-to-SQL with Self-Correction.
>
> [5] Dong et al. 2023. C3: Zero-shot Text-to-SQL with ChatGPT.

---

### Meta-Review · Area_Chair_7ZK1 · 2023-09-11

**Recommendation:** 4

**Metareview:**

There is a consensus among the reviewers that the proposed text-to-SQL method is worthwhile, the experiments are thorough and the paper is generally well written and clear. The reviewers had concerns on the novelty of this work with respect to the existing techniques (and the subsequent comparison with these) and statistical significance tests. The authors have provided detailed response to these. Also, some questions were raised by the reviewers; the authors are encouraged to clarify those as well in the paper.

---

### Decision · Program_Chairs · 2023-10-07

**Decision:**

Accept-Findings

**Comment:**

There is a consensus among the reviewers that the proposed text-to-SQL method is worthwhile, the experiments are thorough and the paper is generally well written and clear. The reviewers had concerns on the novelty of this work with respect to the existing techniques (and the subsequent comparison with these) and statistical significance tests. The authors have provided detailed response to these. Also, some questions were raised by the reviewers; the authors are encouraged to clarify those as well in the paper.